# An Approach Based on Semantic Relationship Embeddings for Text Classification

Ana Laura Lezama-Sánchez [1,†], Mireya Tovar Vidal [1,*,†] and José A. Reyes-Ortiz [2,*,†]

1   Faculty of Computer Science, Benemerita Universidad Autonoma de Puebla, Puebla 72570, Mexico
2   System Department, Universidad Autonoma Metropolitana, Azcapotzalco 02200, Mexico
*   Correspondence: mireya.tovarvidal@viep.com.mx (M.T.V.); jaro@azc.uam.mx (J.A.R.-O.)
†   These authors contributed equally to this work.

**Abstract:** Semantic relationships between words provide relevant information about the whole idea in the texts. Existing embedding representation models characterize each word as a vector of numbers with a fixed length. These models have been used in tasks involving text classification, such as recommendation and question–answer systems. However, the embedded information provided by semantic relationships has been neglected. Therefore, this paper proposes an approach that involves semantic relationships in embedding models for text classification, which is evaluated. Three embedding models based on semantic relations extracted from Wikipedia are presented and compared with existing word-based models. Our approach considers the following relationships: synonymy, hyponymy, and hyperonymy. They were considered since previous experiments have shown that they provide semantic knowledge. The relationships are extracted from Wikipedia using lexical-syntactic patterns identified in the literature. The extracted relationships are embedded as a vector: synonymy, hyponymy–hyperonymy, and a combination of all relationships. A Convolutional Neural Network using semantic relationship embeddings was trained for text classification. An evaluation was carried out for the proposed relationship embedding configurations and existing word-based models to compare them based on two corpora. The results were obtained with the metrics of precision, accuracy, recall, and $F_1$-measure. The best results for the *20-Newsgroup* corpus were obtained with the hyponymy–hyperonymy embeddings, achieving an accuracy of 0.79. For the *Reuters* corpus, $F_1$-measure and recall of 0.87 were obtained using synonymy–hyponymy–hyperonymy.

**Keywords:** deep learning; semantic relationship embeddings; lexical syntactic patterns; convolutional neural networks; text classification

**MSC:** 68T07; 68T09; 68T10; 68T30; 68T50

## 1. Introduction

Semantic relationships between concepts provide essential information in texts. They can indicate the text category that is being analyzed. In addition, they can be represented in processable structures by automatic text classification algorithms.

Representing words, relationships, context, or any information from texts is part of Natural Language Processing ( NLP) tasks. In general, it has been useful for the computer to understand the data. The vectors, which have an appealing, intuitive interpretation, can be the subject of proper operations such as addition, subtraction, and distance measures. They are used in many Machine Learning (ML) algorithms, strategies, and deep learning [1]. Word embeddings have emerged as a topic of research widely used in recent years. They can be used as features in NLP tasks to encode syntactic and semantic word relationships. Other ways of creating embeddings have surfaced, which rely not on neural networks and embedding layers but on leveraging word-context matrices to arrive at vector representations for words [1]. Some models are *GloVe* [2] and *fastText* [3] models. The

*fastText* model has improved over the skip-gram model from [4]. The model learns n-gram embeddings that can be composed to form words. The rationale behind the methodology is that languages that rely heavily on morphology and compositional word-building, such as Turkish and Finnish. These highly inflectional languages have some information encoded in the word parts, which can be used to help generalize unseen words. The *GloVe* model represents ratios of co-occurrences rather than raw counts. The model encodes semantic information about pair of words. *GloVe* is used to derive a suitable loss function for a log-linear model, which is then trained to maximize the similarity of every word pair. *fastText* and *GloVe* explore word-based embeddings, but the relevant information that can provide semantic relationships has been neglected. Although, in the previous works as [5,6], it is proven that semantic relationships such as synonymy, hyponymy, and hyperonymy have provided crucial semantic information. Therefore, semantic relationship-based embeddings can be helpful in several NLP tasks, such as text classification.

This paper presents a novel approach based on relationships extracted from Wikipedia to create embedding models. The creation of embedding models is conditional on the available semantic relations in texts. The process focuses on extracting semantic relationships from an English corpus from Wikipedia, which consists of 5,881,000 documents. Synthetic synonymy, hyponymy, and hyperonymy relationships are extracted with a set of lexical-syntactic patterns created from the literature. The relationships are embedded using the procedure proposed by [7], which is based on matrix factorization. A text classification using CNN was carried out to compare the performance of the relationships-based embeddings proposed in this work and the word-based models as *fasText, GloVe,* and the WordNet-based model presented in [7]. The main contributions of this work are (a) an approach based on semantic relationship embeddings validated in text classification; (b) a comparison of the performance of the semantic relationship embeddings with word-based models; (c) three semantic relationship embedding models that can be useful for NLP applications. It is observed that the results obtained are promising using CNN; nevertheless, they can be variable because each proposed relationship embedding has diverse semantic information.

The rest of this paper is organized as follows. In Section 2, relevant concepts to support this research are presented. In Section 3, the related work to this research is exposed. Section 4 shows the methodology proposed in this research, while the results are presented in Section 5. The conclusions and future work are presented in Section 6. Finally, the references consulted in the development of this work are shown.

## 2. Background Concepts

In this Section, we introduce the relevant concepts to support the research presented in this paper. They are Text Classification, Natural Language Processing, and Deep Learning.

Text Classification begins when a computer system needs to provide a user with the information required quickly and accurately from essays, research reports, medical diagnoses, social media, or news [8]. A system that works with large amounts of documents requires appropriate methods or algorithms for the computer to understand and generate the desired results [9].

The study of the meaning of words and how they are related is a task of Natural Language Processing ( NLP). The NLP has four levels of human language study, one of them is the semantic level. The objective is to discover associations between words that will allow us to define the implicit meaning of each sentence word by word and are used in the same context to give a complete and coherent idea. The associations between the importance of each word are known as semantic relationships. The most used semantic relationships are synonymy, hyponymy, and hyperonymy [9], and their purpose is to provide a clear idea of a sentence. Semantic relations of synonymy are those where there is a relation between two or more words that have the same or almost the same meaning [10]. Hyponymy is a relationship that includes the semantics of one term in another. Hyperonymy is the inverse relation to hyponymy. Therefore, hyperonymy is

the relation of a term that encompasses others semantically [11]. Some existing methods in the literature for extracting synonymy are related to identifying keyphrases where the relevant words of each document are recognized. Then the relationship around them is identified [10].

On the other hand, the literature also uses Convolutional Neural Networks (CNN) that are trained with characteristics of the existing relationships between extracted keyphrases [10]. Lexical-syntactic patterns are generalized linguistic structures or schemes validated by humans that indicate semantic relationships between concepts. The patterns can be applied to identify formalized concepts and semantic relationships in natural language texts [11]. Some methods can extract hyponym–hyperonym and synonymy semantic relationships from a text. The dictionary-based method is based on the use of lexical ontologies such as WordNet [11]. Clustering methods are incorporated to extract this kind of relationship under the premise that similar words share similar contexts [11]. As in synonymy relationships, there are lexical-syntactic patterns validated by experts. Their function will be to strictly extract pairs of words where there is a hyponym–hyperonym relationship [11]. In [7], they use the relations contained in the WordNet lexical database, which has more than 120,000 related concepts. The existing semantic relationships are more than 25 between more than 155,000 words or lemmas, categorized as nouns, verbs, adjectives, and adverbs [12].

So [7] generated a relationship embedding model based on matrix factorization by extracting existing relationships from the WordNet lexical database. An embedding model is a valuable word representation capable of capturing lexical semantics and trained with natural language corpora. These are an improvement over traditional encodings such as Bag-of-Words or the heavyweight *tf-idf*. In recent years they have been included in the use of algorithms developed in NLP [1]. They are reported in the literature as an essential tool in *NLP* tasks such as part-of-speech tagging, chunking, named entity recognition, semantic role tagging, and [1] parsing.

Natural language processing is responsible for generating algorithms so that a computer understands the task it has to perform, imitating human capacity. Some of the more popular embedding models are *word2Vec* [13], *GloVe* [2], *BERT* [14], and *fastText* [3]. The concept of embedding or word embedding model came to fruition in 2013 when Tomas Mikolov and his team at Google developed the embedding model they named *word2vec*. The model has the sub-models continuous Bag of Words (*CBOW* [15]) and skip-gram [4]. *CBOW* receives a context and predicts a target word [15]. On the other hand *skipgram* [4], where each word is represented as a bag of *n*-grams of [2] characters. The *GloVe* embedding model was developed in 2014 by Jeffrey Pennington [13]. The *GloVe* model combines the advantages of the two main family models in the literature: global matrix factorization and local context window methods. The model works with the non-zero elements in a word-word co-occurrence matrix rather than the entire sparse matrix or separate context windows in a large [13] corpus. However, in 2015 Facebook researchers created the embedding model called *fastText*, which has pre-trained models for 294 languages. The authors relied on the *skipgram* [3] model. In 2018, BERT (Bidirectional Encoder Representations from Transformers). BERT is designed to pre-train deep bidirectional representations from the unlabeled text by jointly conditioning the left and right context in all layers [14].

The classification algorithms use word embedding models such as *GloVe* or *fastText*, intending to improve the accuracy of the NLP algorithms. The advancement of technology has made it possible to speed up processes, for example: searching for a specific document, generating a summary, and extracting keyphrases from a text. However, computational approaches need to model knowledge to generate an accurate result as the human being would do [9].

Text Classification is a task carried out by a neural network or an algorithm such as decision trees or nearest neighbors so that large amounts of unordered documents are ordered into classes according to the characteristics of each one [9]. The support vector machine (*SVM*) classifier is used in [16] for adding the land-use types in an irregular land-

use parcel. However, despite the dataset being an image set, each feature was treated as words and the images as documents.

In addition, Text Classification can be carried out with deep learning techniques. Nevertheless, using a deep learning model for text classification involves using a GPU to perform training. The data sets must have an expert assigned manually, which becomes tedious and time-consuming. The technology that supports deep learning and the libraries that allow these techniques to be implemented are evolving rapidly, so it is necessary to be aware of the documentation. The versions used and apply the corresponding updates. On the other hand, a significant advantage of deep learning is that the results obtained are more precise than those provided by a traditional classifier.

Deep learning is a process that can be carried out with Convolutional Neural Networks ( CNN)that have been adopted for text classification tasks, generating successful results. A CNN is a multilayer or hierarchical network and is a high-level feature-based method [16]. CNN is built by stacking multiple layers of features. One layer is made up of $K$ linear filters and an activation function [17]. A CNN is distinguished by the fact that the network weights are shared between different neurons in the [17] hidden layers. Each neuron in the network first computes a weighted linear combination of its inputs. It can be visualized as evaluating a linear filter on the input values [18]. A CNN is the most effective learning to a set of filters. The same set of filters is used on the data set, forcing the network to learn a general encoding or representation of the data. The weights are restricted to be equal across different neurons on the CNN, allowing a better network generalization to perform normalization. What distinguishes a CNN is the presence of a subsampling or pooling layer. The latter allows optimizing the calculation processes to reduce the size of the data in learning new data, allowing for recognition of different characteristics [17].

## 3. Related Works

This section presents related works in the same field. Most use word embedding models such as *GloVe* [2], *fastText* [3] and *word2vec* [13].

Authors such as [7] proposed developing an embedding model based on the WordNet semantic network. The relationships were taken into a relationship matrix, interpreting each relationship with different weights. Subsequently, they applied matrix factorization that included processes such as Pointwise Mutual Information (*PMI*) [19], $L2$ norm, and Principal Component Analysis (*PCA*) [20]. The authors evaluated the performance of the resulting embeddings in a conventional semantic similarity task, obtaining results substantially superior to the performance of word embeddings based on huge data.

In [21], they expose a text classification method that uses the Bag-of-Words representation model with term frequency-inverse document frequency (*tf-idf*) to select the word(s) with the largest sum *tf-idf* as the most representative with similar signification. Furthermore, the *GloVe* word embedding model finds words with similar semantic meanings. The results were compared with methods such as Principal Component Analysis (*PCA*), Linear Discriminant Analysis (*LDA*), Latent Semantic Indexing (*LSI*), a hybrid approach based on *PCA + LDA* with the Naïve Bayes classifier. The data sets were BBC, Classic, and *20-Newsgroup*. The final results showed the proposed algorithm had better classification than the dimension reduction techniques. The authors defined a new metric to evaluate the classifier's performance on reduced features.

Random Multimodel Deep Learning (RMDL) for image, video, symbol, and text classification is proposed by [22]. RMDL aims to find a deep learning structure and architecture by improving robustness and accuracy. The data sets used were MNIST, CIFAR-10, WOS, IMDB, *Reuters*, and *20-Newsgroup*. The text classification techniques used as a reference to evaluate the proposed model are Recurrent Neural Networks (RNN), Convolutional Neural Networks (CNN), and Deep Neural Networks (DNN). In addition, they incorporate the techniques of Support Vector Machine (SVM), Naïve Bayes Classification (NBC), and, finally, Hierarchical Deep Learning for Text Classification (HDLTex). Feature extraction

from texts was performed with the *GloVe* and *word2vec* embedding models. The evaluation metrics used were precision, recall, and $F_1$-measure.

The authors [23] expose an improved model based on Graph Neural Network (GNN) for document classification. The model builds different graphs for each text it receives and then classifies them, reducing memory consumption in a neural network. The data sets were from the *Reuters* and *20-Newsgroup*. The GloVe embedding model was used with a Convolutional Neural Network and Long Short-Term Memory (LSTM). The metric used for model evaluation is accuracy. The results showed that the proposed model achieves higher accuracy than existing literature models.

In [24], a study that compares the accuracy levels of the *word2Vec, GloVe,* and *fastText* embedding models in text classification using a Convolutional Neural Network is carried out. The data sets used in the experiments comprised the UCI KDD file, which contains 19,977 news items and is grouped into 20 topics. The results showed that *fastText* performed better in the classification task. However, when comparing the effects of *GloVe* and *word2Vec* with those provided by *fastText*, the difference in accuracy is not crucially significant, so the authors conclude that their use depends on the data set used. The metric for the evaluation of the proposed model was accuracy.

In [25], a generative probabilistic model for text documents is exposed. The model combines word and knowledge graph embeddings to encode semantic information and related knowledge in a low-dimensional representation. The model encodes each document as points in the von Mises–Fisher distribution. The authors developed a variational Bayesian inference algorithm to learn unsupervised text embeddings. The results showed that the model is applied for text categorization and sentiment analysis. The data sets used were *Obsumed, 20-Newsgroup* and *Reuters*. The evaluation metrics used were $precision, recall, accuracy$, and $F_1$-measure.

The authors [26] present an approach to the problem of classifying texts from sets with few data and sets with data of different lengths. The proposed approach represents texts of any size with 138 features in a fixed-size linguistic vector. The authors addressed two classification tasks: text genres with or without adult content and sentiment analysis. The classification models used were Random Forests, RNN with BiLSTM layer, and the word2vec and BERT models. The evaluation metric used was accuracy.

In [27], the authors compare different strategies for aggregating contextualized word embeddings along lexical, syntactic, or grammatical dimensions. The purpose is to perform semantic retrieval for various natural language processing tasks. The authors defined a set of strategies for aggregating word embeddings along linguistic dimensions. The representations were applied to address tasks such as part-of-speech labeling, identifying relations and semantic frame induction, sequence and word-level labeling, named entity recognition, and word sense disambiguation. The experiments use the *word2vec*, ROBERTA embedding models, and the nearest neighbor classifier. The evaluation metric used was $F_1$-measure. The datasets used were those provided by Semeval 2007, 2010, 2018, CoNLL, SensEval, and TwitterAirline.

In [28], a methodology is presented for sentiment analysis with hybrid embeddings to improve the available pre-trained embedding functions. The authors applied Part of Speech (POS) tagging and the word2position vector over fastText to develop the hybrid embeddings. The metric used in the evaluation process was the accuracy with different deep learning ensemble models and standard sentiment datasets. The data set used was a movie review (MVR). The embedding models used were word2Vec, fastText, and GloVe. The results demonstrate that the proposed methodology is effective for sentiment analysis and can incorporate techniques based on linguistic knowledge to improve the results further.

A text classification model with Convolutional Neural Networks such as Graphical Neural Network (GCN) and Bidirectional Recursive Unit (Bi-GRU) is exposed in [29]. The model was designed to address the lack of ability of neural networks to capture contextual semantics. Furthermore, it extracts complex non-linear spatial features and semantic relationships. The word2vec embedding model is used during the experiments.

The evaluation metrics were precision, recall, and $F_1$-measure. The dataset used in the experiments is THUCNews. The authors report that the proposed model can relate better to the context. Furthermore, by extracting information on spatial features and complex non-linear semantic relationships from the text, the model outperforms other models in terms of accuracy, recall, and $F_1$-measure.

Knowledge graphs as an additional modality for text classification is explored in [30]. Additionally, they explore the inclusion of domain-specific knowledge to deal with domain changes. The authors proved that combining textual embeddings and knowledge graphs achieved good results when applied to a BiLSTM network. The evaluation metrics used were precision, recall, and $F_1$-measure.

The authors in [31] present a study on the text classification task, investigating methods to augment the input to Deep Neural Networks (DNN) with semantic information. Word semantics are extracted from the WordNet lexical database. A vector of semantic frequencies is formed using the weighted concept terms extracted from WordNet. They selected the concepts through various semantic disambiguation techniques, including a basic projection method, a POS-based method, and a semantic embedding method. In addition, they incorporated a weight propagation mechanism that exploits semantic relations and conveys a propagation activation component. The authors incorporated for semantic enrichment the word embedding word2vec, fastText, and GloVe with the proposed semantic vector using concatenation or replacement, and the result was the input of a DNN classifier. The datasets used during the experiments were *20-Newsgroup* and *Reuters*. The evaluation metrics used for evaluation were $F_1$-measure and macro-$F_1$. Experimental results showed that the authors' proposed study increased classification performance.

The authors in [32] propose an investigation on applying a three-layer CNN model in short and long text classification problems through experimentation and analysis. The model is trained using a word embedding model such as fastText. The datasets used are Ag News, Amazon Full and Polarity, Yahoo Question Answer, Yelp Full, and Polarity. In addition, they applied a pre-processing process to each dataset to remove missing, inconsistent and redundant values. Subsequently, each corpus was tokenized and converted into word vectors. The maximum sequence of a sentence was set to the full length of text in the dataset. The authors also applied classifiers such as random forest, logistic regression, extra tree classifier, gradient boosting machine, and stochastic gradient descent. The performance of each classifier was compared with that obtained from the model proposed by the authors. The results obtained showed that the proposed model outperforms traditional classifiers. The evaluation metrics used are precision, recall, accuracy, and $F_1$-measure.

The authors in [33] propose KERMITsystem (Kernelinspired Encoder with Recursive Mechanism for Interpretable Trees). The aim of embed the long symbolic-syntactic history in modern Transformer architecture. The authors aim to investigate whether KERMIT could be used as a meeting point between empiricist and nativist theories exploiting the potential of Transformers models.

The use of dictionary definitions to develop word embeddings for rare words is proposed in [34]. The authors introduce two methods: Definition Neural Network (DefiNNet) and Define BERT (DefBERT). DefiNNet and DefBERT significantly outperform related works and baseline methods devised for producing embeddings of unknown words. DefiNNet significantly outperforms fastText, which implements a method for the same task based on n-grams. Otherhand, DefBERT significantly outperforms the BERT method. Then, the authors concluded definitions in traditional dictionaries helped build word embeddings for rare words.

In this paper, we propose to generate three word embedding models. The models will be based on matrix factorization proposed by [7]. In contrast to [7] the models proposed in this work will be formed by relations extracted with lexical, syntactic patterns from an English Wikipedia corpus. The only additional pre-processing applied over the corpus is to remove non-ASCII characters and convert them to lowercase. To evaluate the performance of the proposed models with the one provided by [7], classification of

the corpus *20-Newsgroup* and *Reuters* will be carried out with a Convolutional Neural Network. The proposed models are evaluated based on *precision*, *accuracy*, *recall* and $F_1$-measure metrics.

## 4. Proposed Approach

This section presents the proposed approach using semantic relationship embeddings for text classification. The approach includes the following process: automatic extraction of semantic relationships from Wikipedia using lexical-syntactic patterns; construction of semantic relationships embeddings as low-dimensional vectors; text classification with a Convolutional Neural Network (CNN); and an evaluation process.

### 4.1. Semantic Relationships Extraction from Wikipedia

Extracting semantic relationships from the English Wikipedia corpus is vital for constructing the proposed embedding models. It is necessary to extract the relations of synonymy, hyponymy, and hyperonymy using lexical-syntactic patterns extracted from the literature for these semantic relationships. Wikipedia is an unlabeled corpus, so the extracted semantic relationships are used for creating embedding models, which will be used for training the CNN algorithm.

This task is carried out as follows: Semantic relations between concepts are extracted from Wikipedia [35] in English. However, Wikipedia is a corpus that lacks labeled datasets with semantic relationships. Therefore lexical-syntactic patterns extracted from the literature are proposed to extract concepts and semantic relations between them. The patterns were converted to regular expressions in the Python programming language. A previous preprocessing was applied to Wikipedia, including removing non-*ascii* characters and converting them to lowercase. The implemented patterns identify semantic relationships (synonymy, hyponymy, and hyperonymy) from Wikipedia.

Each semantic relationship from the literature analyzed the patterns. In this way, pattern sets were obtained for synonymy from [10,18,36] and for hyponymy–hyperonymy from [11,37–42]. For example, some obtained patterns applied in this paper are shown in Tables 1 and 2.

**Table 1.** Lexical-syntactic patterns to extract synonymy relationships.

| Concept 1 | Relation | Concept 2 |
|---|---|---|
| X | *also called* | Y |
| X | *called as* | Y |
| X | *also known as* | Y |
| X | *usually called* | Y |
| X | *is called* | Y |
| X | *are called* | Y |
| X | *sometimes called* | Y |
| X | *know as* | Y |
| X | *also referred to as* | Y |
| X | *often described* | Y |
| X | *commonly known as* | Y |
| X | *also named as* | Y |
| X | *abbreviated as* | Y |
| X | *commonly called as* | Y |
| X | *is often referred to as* | Y |
| X | *is referred to as* | Y |
| X | *alias* | Y |
| X | *aka* | Y |
| X | *as known as* | Y |
| X | *frequently abbreviated as* | Y |
| X | *called as* | Y |
| X | *commonly known as* | Y |
| X | *anciently named as* | Y |

**Table 2.** Lexical-syntactic patterns to extract hyponymy and hyperonymy relationships.

| Concept 1 | Relation | Concept 2 |
|---|---|---|
| *X* | *such as* | *Y* |
| *X* | *include* | *Y* |
| *X* | *especially* | *Y* |
| *X* | *is/are* | *Y* |
| *X* | *is one of the* | *Y* |
| *X* | *like other* | *Y* |
| *X* | *usually* | *Y* |
| *X* | *one of these* | *Y* |
| *X* | *one of those* | *Y* |
| *X* | *be example of* | *Y* |
| *X* | *for example* | *Y* |
| *X* | *which be call* | *Y* |
| *X* | *which be name* | *Y* |
| *X* | *mainly* | *Y* |
| *X* | *mostly* | *Y* |
| *X* | *notably* | *Y* |
| *X* | *particularly* | *Y* |
| *X* | *principally* | *Y* |
| *X* | *in particular* | *Y* |
| *X* | *is a/and/the* | *Y* |
| *X* | *other than* | *Y* |
| *X* | *is the single* | *Y* |
| *X* | *including or/and* | *Y* |
| *X* | *except* | *Y* |
| *X* | *called* | *Y* |
| *X* | *including* | *Y* |
| *X* | *another* | *Y* |
| *X* | *called* | *Y* |
| *X* | **i.e.,** | *Y* |

The patterns are applied to Wikipedia texts to obtain sets of word pairs for each semantic relationship. Each word that composes the semantic relationship is assigned a unique *id*. For example, the relationships *elephant–mammal*, and *cat–animal*, generate the following identifiers: *elephant*:0; *mammal*:1, *cat*:2, and *animal*:3. The assigned identifiers are used to fill a matrix as a traditional representation model, which are will be converted into embedding models as low-dimensional vectors. The number of relationships for synonymy and hyponymy–hyperonymy extracted from Wikipedia is shown in Table 3.

**Table 3.** Semantic relationships extracted.

| Relationship | Total |
|---|---|
| Synonym | 1,200,000 |
| Hyponym–hyperonym | 6,966,042 |

The sets of word pairs for the semantic relationship discovered are used to represent them into embedding models. The Convolutional Neural Network such as deep learning algorithm use the embedded models for text classification.

*4.2. Construction of Semantic Relationship Embeddings*

Each word pair of discovered relationships is assigned a unique identifier for constructing semantic relationship embeddings. Subsequently, a matrix $M$ is filled based on the unique identifiers. The objective is to generate a matrix that represents the semantic relationships and makes up the model to be developed.

The semantic relationship embeddings are based on the identifiers assigned to each relationship. A matrix $M$ is filled by adding a value of 1 to $M$. In Table 4 in position 0,1,

one relationship are represented in the matrix $M$ at position 0,1 add a 1. However *cat* and *mammal* are represented in position 2,1 at Matrix, because *mammal* already has an identifier.

**Table 4.** Example of filling a relationship matrix $M$.

|   | 0 | 1 | 2 | 3 | 4 | 5 | 6 | 7 | 8 |
|---|---|---|---|---|---|---|---|---|---|
| **0** | 0 | 1 | 0 | 0 | 0 | 0 | 0 | 0 | 0 |
| **1** | 1 | 0 | 1 | 0 | 0 | 0 | 0 | 0 | 1 |
| **2** | 0 | 1 | 0 | 1 | 0 | 0 | 0 | 0 | 0 |
| **3** | 0 | 0 | 1 | 0 | 0 | 0 | 1 | 0 | 1 |
| **4** | 0 | 0 | 0 | 0 | 0 | 1 | 0 | 1 | 0 |
| **5** | 0 | 0 | 0 | 1 | 0 | 0 | 0 | 0 | 0 |
| **6** | 0 | 1 | 0 | 0 | 0 | 1 | 0 | 0 | 0 |
| **7** | 0 | 0 | 0 | 1 | 1 | 0 | 0 | 0 | 0 |
| **8** | 0 | 0 | 0 | 1 | 0 | 0 | 0 | 0 | 0 |

Three semantic relationship embeddings are developed from matrix $M$. However, the models include the most frequent relationships from the vocabulary. It was achieved by weighing the type *tf-idf* and selecting the 40,000 most relevant relationships.

For the first embedding configuration, the semantic relationships extract $synonym_1$ and $synonym_2$ relationships. Both synonyms are of interest, adding the relation $synonym_2$ and $synonym_1$. Therefore, in the relationship matrix $M$, a one is assigned to represent the relation $synonym_1$ and $synonym_2$ and $synonym_2$ and $synonym_1$.

The second embedding configuration represents the hyponymy and hyperonymy relationships and also represents the hyperonymy and hyponymy relationships at the same time.

Given the semantic contribution that synonymy, hyponymy, and hyperonymy generate, it is proposed to generate a model with the three semantic relations in a single model. A one is assigned in the $M$ matrix for the three relationships. Therefore, the relationship matrix $M$ is assigned to "1" value that represents $synonym_1$ and $synonym_2$, $synonym_2$ and $synonym_1$, hyponymy and hyperonymy, and hyperonymy and hyponymy, respectively. The number of relationships used in this model was only 50% of those used in the model that only includes synonyms and 50% of those used in the model that only includes hyponymy and hyperonymy.

For each embedding configuration, the $M$ relationship matrix will be generated, i.e., the semantic relationships are represented with a 1. Subsequently, the following procedure is applied:

1. Enrichment of $M$ to represent the strength of the semantic affinity of identified relations or nodes that are not directly connected by an edge, using the equation:

$$M_G = (I - \alpha M) \tag{1}$$

where

   (a) $I$ is the identity matrix.
   (b) $M$ is the array where each $M_{i,j}$ counts the number of paths of length $n$ between nodes $i$ and $j$.
   (c) $\alpha$ decay factor that determines how shorter paths dominate.

2. $M_G$ is subjected to the Pointwise Mutual Information $(PMI)$ [19] to reduce the possible bias introduced by the conversion to words with more senses.

3. For a correct conversion application: each line in $M_G$ is normalized using the $L2$ norm to correspond to a vector whose scores sum to 1, corresponding to a transition matrix.

4. The $M_G$ matrix is transformed using Principal Component Analysis ($PCA$) [20] to reduce the vectors' size and set the dimension of the encoded semantic space to 300.

To evaluate the performance of each proposed model, it is proposed to carry out a classification of two existing data sets in the literature.

### 4.3. Text Classification Using CNN

The objective of classifying texts with a Convolutional Neural Network using the proposed semantic relationship embeddings is to evaluate the performance of each configuration. The three proposed semantic relationship embeddings and the word-based embeddings models are applied individually to classify two corpora. The main aim is to compare the proposed semantic relationship embeddings with GloVe, fastText, and WordNet-based [7] models.

The datasets *20-Newsgroup* and *Reuters* exposed in Section 5.1 are used to evaluate the performance of the embedding models.

The *20-Newsgroup* and *Reuters* sets are preprocessing prior to use in conjunction with embedding vectors in the Convolutional Neural Network. It includes the following steps:

1. Remove html tags;
2. Remove punctuation symbols;
3. Remove stop words;
4. Convert to lowercase;
5. Remove extra whitespace.

The neural network used is composed of an input layer, an intermediate layer and an output layer. The middle layer is composed of:

1. Embedding layer: embedding layer to incorporate a pre-trained embedding model.
2. Cov1D layer: creates a kernel that convolves with the input of the layer over a single dimension to produce an output tensor
3. MaxPooling1D layer: Downsamples the input representation by taking the maximum value over a spatial window of size $n$.
4. Concatenate layer: takes a list of tensors as input, and returns a single tensor
5. Dropout layer: prevents overfitting by giving each neuron a 50% probability of not activating during the training phase.
6. Flatten layer: transforms the shape of the input to a one-dimensional vector.
7. Dense layer: fully connected layer with an output dimensionality of 512 and ReLu activation function.

The classification performance was evaluated with precision, accuracy, recall, and $F_1$-measure metrics.

## 5. Results and Discussion

This section presents the results obtained with the proposed semantic relationship embeddings for text classification using *20-Newsgroup* and *Reuters* corpus. In addition, they are compared with the results obtained with the GloVe, fastText, and WordNet-based models. The proposed models are contrasted with GloVe since it is based on representing proportions of co-occurrences encoding semantic information about a pair of words. On the other hand, the comparison is made with fastText since it learns embeddings of n-grams composed to form words and depends on the morphology and construction of the words considered. In addition, the performance obtained when classifying texts with the embedding model based on WordNet is also exposed. This model is compared since the models proposed in this paper consider semantic relationships such as [7] proposal. They create embedding vectors with words presented in semantic relationships between concepts from WordNet. Unlike this work, semantic relationships between concepts are extracted from Wikipedia.

The results obtained provided a view of the three proposed semantic relationship embeddings. Based on them, it can be seen that they still do not outperform the GloVe

or fastText models. However, they are capable of outperforming the model based on WordNet. The following sections present the results obtained and evaluated with the metrics *precision*, *recall*, *accuracy*, and $F_1$-measure, as well as the datasets used in this work.

### 5.1. Datasets

An English corpus from Wikipedia was used to extract semantic relationships (synonymy, hyponymy, and hyperonymy). The extraction was performed using a repository of lexical-syntactic patterns previously taken from the literature for the three semantic relationships. Each pattern was converted to a regular expression. The extracted semantic relationships are what will form the embedding models. Table 5 exposes the number of documents and tokens of the Wikipedia corpora for the extraction of semantic relationships as well as *Reuters* (https://trec.nist.gov/data/reuters/reuters.html, accessed on 1 May 2020) and *20-Newsgroup* (http://qwone.com/~jason/20Newsgroups/, accessed on 1 May 2020) for the classification task.

**Table 5.** Description of dataset.

| Corpus | Documents | Tokens |
|---|---|---|
| Wikipedia | 5,881,000 | 3,380,578,354 |
| *20-Newsgroup* | 20,000 | 1,800,385 |
| Reuters | 18,456 | 3,435,808 |

Table 6 exposes the semantic relationship embeddings used in this research. The *GloVe* and *fastText* models are the most popular in the literature and have been trained on large corpora. On the other hand, a model based on WordNet with 60,000 tokens used is exposed. The models proposed in this work are also exposed: synonymy and hyponymy-hyperonymy; and a combination of both. As can be seen, the relationships that form these three models contain fewer relationships than those shown in Table 3. The computer equipment used during the experiments has a memory supporting a low number of tokens.

**Table 6.** Embedding models.

| Embedding Models | Data | Vector Size |
|---|---|---|
| GloVe | 6 billion tokens and have representations for 400 thousand words | 300 |
| fastText | 1 million word vectors and 16 billion tokens | 300 |
| WordNet | 60 thousand tokens | 300 |
| Synonyms | 40,000 tokens | 300 |
| Hyponym-Hyperonym | 40,000 tokens | 300 |
| Combination | 40,000 tokens | 300 |

### 5.2. Experimental Results

The results of evaluating the performance of the three proposed semantic relationship embeddings are presented, as well as the GloVe, fastText, and WordNet-based models.

The results showed that the proposed semantic relationship embeddings obtain better results than those proposed with relationships extracted from WordNet [7].

Table 7 shows the results obtained by classifying the corpora *20-Newsgroup* and *Reuters*. The precision metric is identified by the tag *P*, recall by *R*, accuracy by *A*, and $F_1$ measure by the tag $F_1$. It is observed that the results obtained when applying the WordNet-based relationship embedding model do not exceed the results obtained with the *GloVe* and *fastText* models.

Secondly, the results for the corpus *20-Newsgroup* exceed the results obtained with *fastText* with a recall of 0.78 and an accuracy of 0.79 for the model that involves three proposed semantic relationship embeddings.

In addition, it outperforms WordNet, obtaining results of 0.75, 0.78, and 0.79 for the precision, recall, and accuracy metrics, respectively.

The results when classifying the corpus *Reuters* outperforms *GloVe* and *fastText* with an $F_1$ of 0.70 and a recall of 0.74 and only *GloVe* with an accuracy of 0.84 for the model incorporating synonyms. For the same corpus, a performance of 0.80 is obtained for the precision metric and 0.87 for the recall and $F_1$-score metrics with the model incorporating three semantic relationships, improving WordNet.

In addition, the semantic relationship embedding that incorporates synonymy obtains an accuracy of 0.84 in the classification of the corpus *Reuters* versus an accuracy of 0.68 reported by the WordNet-based model.

It is estimated that the results exceeded those obtained with WordNet because the relationships included in each proposed model were the most frequent in the total number of relationships obtained.

In some cases the exposed models outperformed *GloVe* and *fastText*. However, these results are still shallow, so it is expected that including a greater number of semantic relationships in each model will exceed both the model exposed by [7] also *GloVe* and *fastText*.

**Table 7.** Results obtained with the CNN and the proposed models.

| Embedding Model | *20-Newsgroup* | | | | *Reuters* | | | |
|---|---|---|---|---|---|---|---|---|
| | **P** | **R** | **A** | **$F_1$** | **P** | **R** | **A** | **$F_1$** |
| fastText | 0.76 | 0.74 | 0.75 | 0.75 | 0.72 | 0.71 | 0.71 | 0.71 |
| GloVe | 0.79 | 0.79 | 0.79 | 0.79 | 0.72 | 0.66 | 0.66 | 0.67 |
| WordNet | 0.66 | 0.64 | 0.64 | 0.64 | 0.71 | 0.68 | 0.68 | 0.68 |
| Hyponym-hyperonym | 0.75 | 0.78 | 0.79 | 0.66 | 0.72 | 0.67 | 0.67 | 0.68 |
| Synonyms | 0.66 | 0.64 | 0.64 | 0.64 | 0.70 | 0.74 | 0.84 | 0.70 |
| Combination | 0.67 | 0.59 | 0.59 | 0.60 | 0.80 | 0.87 | 0.77 | 0.87 |

## 6. Conclusions and Future Work

This paper has presented an approach for text classification using semantic relationship embeddings and Convolutional Neural Networks as deep learning. The semantic relationship embeddings are compared with fasText, GloVe, and WordNet-based models to evaluate and compare their performance.

Semantic relationships were extracted from Wikipedia using lexical-syntactic patterns. The semantic relationship embeddings presented incorporate synonymy, and hyponymy–hyperonymy, a combination of them. Furthermore, $synonym_1 - synonym_2$ and $synonym_2 - synonym_1$ are included. On the other hand, the inverse of the hyponym–hyperonym is also included. It generates three semantic relationship embeddings: synonyms, hyponyms–hyperonyms, and the three relationships. On the other hand, the behavior of each model presented is evaluated through text classification. In addition, its performance is compared with the results obtained when evaluating the performance of fasText, GloVe, and WordNet-based models. The results showed that the proposed semantic relationship embeddings outperform the WordNet-based models.

The main contributions of this paper are: an approach based on semantic relationship embeddings validated in text classification; the extraction of semantic relationships from Wikipedia in English using lexical-syntactic pattern; The use of synonymy, hyponymy, and hyperonymy as semantic relationships to generate embedding as low-dimensional vectors; a comparison of the performance of the semantic relationship embeddings with word and WordNet-based models;

In this way, results showed the lack of a more significant number of tokens in each model. In addition, three proposed embeddings expose the importance of semantic relationships providing complete ideas in a text, which is helpful for text classification tasks by enriching the vectors for documents. Although the results are not the best in comparison with GloVe and FasText, the approach can be helpful for data analysts because semantic relationship embeddings continue to be a tool that improves results for automatic tasks that involve the treatment of textual information. It is observed that the results obtained are variable because each proposed embedding has different semantic information. Furthermore, the approach has become a helpful resource in the natural language field.

As future work, different models of the lexical-syntactic patterns to extract semantic relationships could be incorporated. As well as adding other semantic relationships such as part-whole or causal and semantic roles, it is considered that it will improve the levels of performance obtained. In addition, an investigation addressing Spanish News and Wikipedia in Spanish will be relevant. Finally, adding Word Embeddings based on BERT model in the experiments to compare the performance with current models.

**Author Contributions:** Funding acquisition, J.A.R.-O.; Investigation, A.L.L.-S., M.T.V., and J.A.R.-O.; Methodology, J.A.R.-O.; Supervision, M.T.V. and J.A.R.-O.; Writing—original draft, A.L.L.-S., M.T.V., and J.A.R.-O.; Writing—review and editing, M.T.V. and J.A.R.-O. All authors have read and agreed to the published version of the manuscript.

**Funding:** This research received no external funding.

**Institutional Review Board Statement:** Not applicable.

**Informed Consent Statement:** Not applicable.

**Data Availability Statement:** Not applicable.

**Acknowledgments:** The authors would like to thank Universidad Autonoma Metropolitana, Azcapotzalco. The present work has been funded by the research project SI001-18 at UAM Azcapotzalco and by the Consejo Nacional de Ciencia y Tecnologia (CONACYT) with the scholarship number 788155. The authors thankfully acknowledge computer resources, technical advice, and support provided by Laboratorio Nacional de Supercómputo del Sureste de México (LNS), a member of the CONACYT national laboratories, with project No 202103090C and partly by project VIEP 2021 at BUAP.

**Conflicts of Interest:** The authors declare no conflicts of interest.

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
