# Peer review of "An Approach Based on Semantic Relationship Embeddings for Text Classification"

_mathematics, doi:10.3390/math10214161_

Round 1
Reviewer 1 Report
1. The summary of the existing methods in the introduction is confusing, it’s a better way to divide the methods into several groups and tell readers the pros and cons of each classification method.
2. It’s better to list the major contribution of your work at the end of the introduction section.
3. It’s better to present the references of the PMI and PCA for the readers.
4. The main contribution of the paper is not clear and outstanding, the methodological part does not show specific innovation, and the experimental results are rather general. You may need to reorganize the paper to highlight the contribution and importance of the work.
Author Response
Point 1: The summary of the existing methods in the introduction is confusing, it's a better way to divide the methods into several groups and tell readers the pros and cons of each classification method.
Response 1:
A Background Concepts section has been added where we present relevant concepts to support the research presented in this article. They are text classification, natural language processing, and deep learning. To more clearly list the contributions of the work, the introduction section was restructured and before presenting the distribution of the work the main contributions of this work are presented.
Section 3 is restructured, carrying out a detailed description of the proposed method and hand in hand with the updated title the contribution made is reflected. The state of the art is also updated, which includes works that use BERT. The state of the art is also updated, which includes works that use BERT.
Point 2: It's better to list the major contribution of your work at the end of the introduction section.
Response 2: To more clearly list the contribution of the work, the introduction section was restructured and before presenting the distribution of the work, the following paragraph was added: The main contributions of this work are a) an approach based on semantic relationship embeddings validated in text classification ; b) a comparison of the performance of the semantic relationship embeddings with word-based models; c) three semantic relationship embedding models that can be useful for NLP applications. It is observed that the results obtained are promising using CNN, however, they can be variable because each proposed relationship embedding has diverse semantic information. The main contributions of this work are a) an approach based on semantic relationship embeddings validated in text classification; b) a comparison of the performance of the semantic relationship embeddings with word-based models; c) three semantic relationship embedding models that can be useful for NLP applications. It is observed that the results obtained are promising using CNN, however, they can be variable because each proposed relationship embedding has diverse semantic information.
Point 3: It's better to present the references of the PMI and PCA for the readers.
Response 3: References 34 and 35 are added for PMI and PCA respectively. They are located on page 4.
Point 4: The main contribution of the paper is not clear and outstanding, the methodological part does not show specific innovation, and the experimental results are rather general. You may need to reorganize the paper to highlight the contribution and importance of the work.
Response 4: The title, abstract, and introduction are updated. The abstract describes the problem being addressed. The introduction clearly describes the contribution of the work now it is an approach that does not consider the models but now it is an approach that uses semantic relations of type synonymy, hyponymy and hyperonymy with the aim of covering the neglect of information embedded by semantic relations, i.e. contribution would be Semantic Relationship Embeddings. The title has been changed to focus on the contribution of the work. The original title was Semantic Relationship-based Embedding Models for Text Classification. The new title are An Approach based on Semantic Relationship Embeddings for Text Classification.
Reviewer 2 Report
In this manuscript, the authors propose embedding representation models for text classification. State of the art, there are plenty of models to solve this problem. They go from syntax to semantics and lexicon. In this paper, the authors use semantic relations. These relations are words that are related to each other and provide a complete idea of a text.
The authors, throughout the paper, make a reasonably strong claim (I did not say it is wrong). The authors say that an embedding model involving semantic relations provides better performance for tasks that use them.
Building on this solid claim in this paper, three embedding models are presented based on semantic relations extracted from Wikipedia to classify texts.
The semantic relations considered are synonyms, hyponyms, and hyperonym.
The evaluations were done on known corpora such as 20-Newsgroup and Reuters.
The measures considered are the classical ones.
The work seems fluent and there are no particular difficulties in reading but there are some considerations to be made:
1) the introduction is really very long, and the essential part ("This research aims to develop") comes very late.
2) The Related Works part definitely mentions a firm leader of modern NLP, BERT. PEr why didn't you propose a model of the Transformer family?
3) Some works exploit BERT; see DefBERT in "Lacking the embedding of a word? look it up into a traditional dictionary" and get good performances. I recommend that you also review it in the background.
4) The proposed model is a CNN. I suggest you focus less on Table 2,3,4 and describe your idea better.
5) The results section is weak. First thing, the results table should have bold fonts. Second I think the results should be discussed better.
I strongly recommend that you consider the work "Dis-Cover AI Minds to Preserve Human Knowledge," which is a cornerstone in understanding what is happening in machine learning and how statistical learners "try" to represent the semantic, syntactic and symbolic world.
Author Response
Point 1: the introduction is really very long, and the essential part ("This research aims to develop") comes very late.
Response 1: In response to this comment, the following actions were carried out:
The title, abstract, and introduction are updated. The abstract describes the problem being addressed. The introduction clearly describes the contribution of the work now it is an approach that does not consider the models but now it is an approach that uses semantic relations of type synonymy, hyponymy and hyperonymy with the aim of covering the neglect of information embedded by semantic relations, i.e. contribution would be Semantic Relationship Embeddings. The title has been changed to focus on the contribution of the work. The original title was Semantic Relationship-based Embedding Models for Text Classification. The new title is An Approach based on Semantic Relationship Embeddings for Text Classification.
A Background Concepts section has been added where we present relevant concepts to support the research presented in this article. They are text classification, natural language processing, and deep learning. To more clearly list the contributions of the work, the introduction section was restructured and before presenting the distribution of the work the main contributions of this work are presented.
The state of the art is also updated, which includes works that use BERT.
Point 2: The Related Works part definitely mentions a firm leader of modern NLP, BERT. PEr why didn't you propose a model of the Transformer family?
Response 2:
The objective was to develop an embedding model based on semantic relations like the one exposed in [1]. The difference was only in the origin of the semantic relationships, so using a model of the Transformer family would not meet the initial objective.
Point 3: Some works exploit BERT; see DefBERT in "Lacking the embedding of a word? look it up into a traditional dictionary" and get good performances. I recommend that you also review it in the background.
Response 3: In response to this recommendation, the article is read and integrated into the state of the art. In addition, it is considered to integrate BERT in future works.
Point 4: The proposed model is a CNN. I suggest you focus less on Table 2,3,4 and describe your idea better.
Response 4:
Section 3 is restructured, carrying out a detailed description of the proposed method and table 3 is eliminated. In addition, a section on background concepts is added in order to make the information in the introduction clearer. Which turns section 3 into section 4.
Point 5: The results section is weak. First thing, the results table should have bold fonts. Second I think the results should be discussed better.
Response 5:
In response to this comment, the following actions were carried out: Bold fonts added in all tables and results section is restructured with the description more large and therefore with the changes made it is more related and the contribution made is reflected proposed, its usefulness and novelty.
Point 6: I strongly recommend that you consider the work "Dis-Cover AI Minds to Preserve Human Knowledge," which is a cornerstone in understanding what is happening in machine learning and how statistical learners "try" to represent the semantic, syntactic and symbolic world.
Response 6:
In response to this comment, the following actions were carried out: Read the article and integrate into the state of the art.
[1]C. Saedi, A. Branco, J. A. Rodrigues, and J. R. Silva. Wordnet Embeddings. In Proceedings of the 502 third workshop on representation learning for NLP 2018, 122–131.
Reviewer 3 Report
Research related to text classification is important and timely.
The use of Convolutional Neural Networks (CNN) often gives good results in text analysis, but it is not possible to analyze subsequent decision steps as, for example, in decision trees. In this paper, it is not clear the novelty of the research since a comparison with already existing methodologies is missing.
Generally, the work is written correctly, the research and research conclusions are correct. I have no major comments, but the description of the article organization needs to be improved (there is section 8 there, which is not included in this paper).
Author Response
Point 1: The use of Convolutional Neural Networks (CNN) often gives good results in text analysis, but it is not possible to analyze subsequent decision steps as, for example, in decision trees. In this paper, it is not clear the novelty of the research since a comparison with already existing methodologies is missing.
Response 1: In response to this comment, the following actions were carried out:
The title, abstract, and introduction are updated. The abstract describes the problem being addressed. The introduction clearly describes the contribution of the work now it is an approach that does not consider the models but now it is an approach that uses semantic relations of type synonymy, hyponymy and hyperonymy with the aim of covering the neglect of information embedded by semantic relations, i.e. contribution would be Semantic Relationship Embeddings. The title has been changed to focus on the contribution of the work. The original title was Semantic Relationship-based Embedding Models for Text Classification. The new title is An Approach based on Semantic Relationship Embeddings for Text Classification.
A Background Concepts section has been added where we present relevant concepts to support the research presented in this article. They are text classification, natural language processing, and deep learning. To more clearly list the contributions of the work, the introduction section was restructured and before presenting the distribution of the work the main contributions of this work are presented.
Section 3 is restructured, carrying out a detailed description of the proposed method and hand in hand with the updated title the contribution made is reflected. The state of the art is also updated, which includes works that use BERT.
Point 2: Generally, the work is written correctly, the research and research conclusions are correct. I have no major comments, but the description of the article organization needs to be improved (there is section 8 there, which is not included in this paper).
Response 2:
The label of the results section is corrected in the description of the distribution of work.
Round 2
Reviewer 1 Report
The manuscript has been improved according to the reviews' comments, there's remaining one minor comment:
The text classification techniques used as a reference to evaluate the proposed model are Recurrent Neural Networks (RNN), Convolutional Neural Networks (CNN), and Deep Neural Networks (DNN). In addition, they incorporate the techniques of Support Vector Machine (SVM)....
To improve the literature review of this paper, the related state-of-the-art methods combining CNN, BoW and SVM should be cited: https://doi.org/10.1016/j.rse.2022.112916
Author Response
The article: https://doi.org/10.1016/j.rse.2022.112916 was revised and cited for the text classification task. The article was added to Section 2 in "Background concepts" on page 3, lines 138-141. In addition, in the same section on page 4, lines 152-153, a paragraph with a citation is added.
And the reference:
Zhu, Q.; Lei, Y.; Sun, X.; Guan, Q.; Zhong, Y.; Zhang, L.; Li, D. Knowledge-guided land pattern depiction for urban land use mapping: A case study of Chinese cities. Remote Sensing of Environment, 2022, Volume 272, 112916.